# Phycobiliprotein Peptide Extracts from *Arthrospira platensis* Ameliorate Nonalcoholic Fatty Liver Disease by Modulating Hepatic Lipid Profile and Strengthening Fat Mobilization

**DOI:** 10.3390/nu15214573

**Published:** 2023-10-27

**Authors:** Jing Liu, Huan Wu, Yan Zhang, Changbao Hu, Dongyu Zhen, Pengcheng Fu, Yanfu He

**Affiliations:** 1International School of Public Health and One Health, Hainan Medical University, Haikou 571199, China; csuliujing@163.com; 2State Key Laboratory of Marine Resource Utilization in South China Sea, Hainan University, Haikou 570228, China; 19989767254@163.com (H.W.); zy980618@163.com (Y.Z.); 3School of Food Science and Engineering, Hainan University, Haikou 570228, China; 18189830072@163.com (C.H.); zhendy2001@163.com (D.Z.); 4Hainan Provincial Engineering Research Centre of Aquatic Resources Efficient Utilization in the South China Sea, Hainan University, Haikou 570228, China

**Keywords:** bioactive peptides, nonalcoholic fatty liver disease, lipidomic, fat mobilization

## Abstract

*Arthrospira platensis* phycobiliprotein peptide extracts (PPEs) exhibit potential mitigative effects on hepatic steatosis. However, the precise role of PPEs in addressing high-fat-induced nonalcoholic fatty liver disease (NAFLD), as well as the underlying mechanism, remains to be elucidated. In this study, NAFLD was induced in rats through a high-fat diet (HFD), and the rats were subsequently treated with PPEs for a duration of 10 weeks. The outcomes of this investigation demonstrate that PPE supplementation leads to a reduction in body weight gain, a decrease in the accumulation of lipid droplets within the liver tissues, alterations in hepatic lipid profile, regulation of lipolysis-related gene expression within white adipose tissues and modulation of intestinal metabolites. Notably, PPE supplementation exhibits a potential to alleviate liver damage by manipulating neutral lipid metabolism and phospholipid metabolism. Additionally, PPEs appear to enhance fat mobilization by up-regulating the gene expression levels of key factors such as *HSL*, *TGL*, *UCP1* and *UCP2*. Furthermore, PPEs impact intestinal metabolites by reducing the levels of long-chain fatty acids while concurrently increasing the levels of short-chain fatty acids. The findings from this study unveil the potential of PPE intervention in ameliorating NAFLD through the modulation of hepatic lipid profile and the reinforcement of the fat mobilization of intestinal metabolites. Thus, PPEs exhibit noteworthy therapeutic effects in the context of NAFLD.

## 1. Introduction

Nonalcoholic fatty liver disease (NAFLD) has emerged as a progressively common and impactful chronic liver disease, primarily due to its strong correlation with a preference for fatty foods and metabolic syndrome, compounded by the absence of effective pharmacological therapies [1]. The spectrum of NAFLD encompasses simple steatosis, hepatic steatohepatitis and cirrhosis, with implications ranging from heightened liver enzymes to cryptogenic cirrhosis and even hepatocellular carcinoma [2]. Prolonged adherence to a high-fat diet leads to the accumulation of visceral adipose tissue, generating a cascade of signals capable of disrupting cellular metabolism, and ultimately causes hepatic steatosis and fosters a proinflammatory environment [3]. A cascade of detrimental processes including oxidative stress response, pathways related to unfolded protein responses, lipotoxicity and apoptotic pathways collectively contribute to liver fibrosis. Over time, these processes can escalate to liver cirrhosis and potentially culminate in hepatocellular carcinoma [4]. Therefore, a crucial strategy for mitigating NAFLD and staving off severe liver disorders involves bolstering lipid export from the liver and promoting the catabolism of fatty acids.

Notably, a form of NAFLD triggered by prolonged high-fat diets often associated with liver aging can be ameliorated by a combined exercise and dietary intervention that stimulates lipophagy [5]. From a mechanistic perspective, the excessive accumulation of fat in NAFLD is predominantly attributed to disruptions in lipid metabolism homeostasis. This includes an upsurge in lipid uptake and biosynthesis, concomitant with a reduction in lipid degradation [6]. *Arthrospira platensis,* an edible cyanobacterial species with a millennia-long history of consumption, has generated interest as a nutritional source [7]. Initially, the focus on *Arthrospira platensis* centered on its protein-rich content, particularly the phycobiliprotein, in response to the escalating global demand for protein by the world population [8]. The amino acid profile of phycobiliprotein is comprehensive and has a balanced essential amino acid composition. Recently, phycobiliprotein peptides have drawn much attention for their therapeutic value in alleviating obesity, hyperglycemia, hyperlipidemia, and cardiovascular disease and enhancing immunity. These therapeutic attributes are rooted in their diverse biological properties, as documented in the literature [9]. However, the specific in vivo amelioration effect and underlying mechanisms through which *Arthrospira platensis* phycobiliprotein peptides exert their impact on high-fat diet (HFD)-induced NAFLD remain unexplored. Further research is needed to shed light on these aspects and to elucidate the potential therapeutic role of these peptides.

The aim of this study is to assess the potential therapeutic impact of *Arthrospira platensis* phycobiliprotein peptide extracts (PPEs) on NAFLD and to delve into the underlying mechanisms through the lens of lipid metabolism. Through this research endeavor, we aim to ascertain whether PPEs can ameliorate NAFLD and to provide initial insights into the mechanisms responsible for this amelioration. Specifically, our investigation entails the examination of PPEs’ effects on hepatoprotection within a rat model of NAFLD induced by HFD. Our research underscores that PPEs yield a hepatoprotective effect by rectifying a disrupted lipid metabolism homeostasis and augmenting the mobilization of fats. These findings contribute empirical support to the notion that PPEs might constitute an innovative strategy for dietary intervention aimed at managing NAFLD and its associated manifestations.

## 2. Materials and Methods

### 2.1. Preparation of PPE

*Arthrospira platensis* phycobiliprotein were generously provided by Xindaze Biotech Co., Ltd. (Fuqing, China). Enzymatic hydrolysis of these phycobiliprotein resulted in the production of *Arthrospira platensis* PPEs. The detail preparation method of PPEs was executed in accordance with our previous study [10]. In brief, enzymolysis and subsequent centrifugation steps were carried out, leading to the formation of a PPE, which constitutes a mixture of small molecule peptides (2 to 15 amino acids). The specific peptide sequences are outlined in Appendix A.

### 2.2. Animal Experiments and Sample Collection

A total of 24 male Sprague Dawley (SD) rats, aged 6 weeks and weighing approximately 190 ± 10 g, were procured from Hunan SJA Laboratory Animal Co., Ltd. (Changsha, China). Upon arrival, the rats were housed individually in ventilated plastic cages (6 rats per cage) under standard laboratory conditions, maintaining a temperature 23 ± 1 °C and a 12 h daylight cycle. The rats were provided with unrestricted access to a standard full value chow diet and water. Ethical clearance for all animal procedures was granted by the Hainan University Institutional Animal Welfare and Ethical Committee (HNDX2021076). Two distinct diets were used in the study, both obtained from Vital River Laboratory Animal Co., Ltd. (Beijing, China). The high-fat diet (referred to as DIO, H10045) consisted of 45% fat, 20% protein and 35% carbohydrates. The chow diet (referred as to DIO, H10010) consisted of 10% fat, 20% protein and 70% carbohydrates. Following a week of acclimation, the rats were randomly divided into four groups (*n* = 6), including the chow diet control group (Chow Control), high-fat diet control group (HFD Control), high-fat diet with low-dose PPE group (PPE_L, administered 125 mg/(kg·BW·d)) and high-fat diet with high-dose PPE group (PPE_H, administered 500 mg/(kg·BW·d)). After 6 weeks of being fed a high-fat diet, the rats in the PPE_L and PPE_H groups were subjected to low- or high-dose PPE treatment via intragastric gavage, respectively, while the Chow Control and HFD Control groups were administered an equal volume of sterile water once a day for 4 weeks. Weekly body weight measurements were taken, following overnight fasting. After the anesthesia, the blood, white adipose tissue, liver tissue and cecal contents of the rats were collected for subsequent experiments. Serum was obtained by centrifuging the blood at 4 °C, 3000 rpm for 20 min, and then stored at −80 °C. Liver tissue was partitioned into two portions: one was fixed in 4% paraformaldehyde for lipid droplet analysis, and the remaining portion was stored at −80 °C. White adipose tissue and cecal contents were also stored at −80 °C in an ultra-low temperature freezer.

### 2.3. Serum Biochemical Assessment

The serum levels of triglyceride (TG), total cholesterol (TC), low-density lipoprotein cholesterol (LDL-C) and high-density lipoprotein cholesterol (HDL-C) were quantified using biochemical kits obtained from Jiancheng (Nanjing, China). The procedures specified in the kit instructions were meticulously followed.

### 2.4. Quantification of Lipid Abundance in Liver Tissues by Oil Red O

The liver tissues were fixed in 4% paraformaldehyde for more than 24 h, followed by dehydration through immersion in 20% to 30% sucrose until complete. The tissues were then embedded in paraffin and sectioned into slices of 10 μm thickness; these slices were subsequently stained with oil red O, and observations were made using a microscope. The quantification of lipid abundance in liver tissues was conducted using Image J 1.8.0 software.

### 2.5. Lipidomic Analysis

For lipidomic analysis, 50 mg of liver tissue was accurately weighed. Metabolites were extracted using a solution composed of methanol: water (2:5, *v*/*v*) containing 0.02 mg/mL L-2-chlorophenylalanin as an internal standard. The mixture was allowed to settle at −10 °C and subsequently treated using a high-throughput tissue crusher Wonbio-96c (Shanghai Wanbo Biotechnology Co., Ltd., Shanghai, China) at 50 Hz for 6 min; ultrasound treatment at 40 kHz for 30 min at 5 °C followed this step. The samples were placed at −20 °C for 30 min to precipitate proteins. After centrifugation at 13,000× *g* at 4 °C for 15 min, the supernatant was meticulously transferred to sample vials for LC-MS/MS analysis. As a part of the system conditioning and quality control, a pooled quality control sample (QC) was generated by mixing equal volumes of all individual samples. The QC samples were treated and tested in the same protocol as the actual samples.

The UHPLC-Q Exactive HF-X system from Thermo Fisher Scientific (Waltham, MA, USA) was employed for this analysis. In total, 2 μL of the sample was separated using an HSS T3 column (100 mm × 2.1 mm i.d., 1.8 μm) before mass spectrometry detection. The mobile phases consisted of 0.1% formic acid in water/acetonitrile (95:5, *v*/*v*) (solvent A) and 0.1% formic acid in acetonitrile/isopropanol/water (47.5:47.5:5, *v*/*v*) (solvent B). The solvent gradient changed according to the following conditions: from 0 to 3.5 min, 0% B to 24.5% B (0.4 mL/min); from 3.5 to 5 min, 24.5% B to 65% B (0.4 mL/min); from 5 to 5.5 min, 65% B to 100% B (0.4 mL/min); from 5.5 to 7.4 min, 100% B to 100% B (0.4 mL/min to 0.6 mL/min); from 7.4 to 7.6 min, 100% B to 51.5% B (0.6 mL/min); from 7.6 to 7.8 min, 51.5% B to 0% B (0.6 mL/min to 0.5 mL/min); from 7.8 to 9 min, 0% B to 0% B (0.5 mL/min to 0.4 mL/min); and from 9 to 10 min, 0% B to 0% B (0.4 mL/min) for equilibrium. The flow rate was set to 0.4 mL/min. The column temperature was maintained at 40 °C. The mass spectrometric data were collected in either positive or negative ion mode. The optimal conditions were set as follows: heater temperature, 425 °C; capillary temperature, 325 °C; sheath gas flow rate, 50 arb; Aux gas flow rate, 13 arb; ion-spray voltage floating, −3500 V in negative mode and 3500 V in positive mode, respectively; normalized collision energy, 20–40–60 V rolling for MS/MS. Full MS resolution was 60,000, and MS/MS resolution was 7500. Data acquisition was performed in the data-dependent acquisition mode. The detection was carried out within a mass range of 70–1050 *m*/*z*.

Upon completion of mass spectrometry detection, the raw data underwent preprocessing using Progenesis QI 2.0 (Waters Corporation, Milford, CT, USA) software. A three-dimensional data matrix in CSV format was exported, encompassing sample information, metabolite name and mass spectral response intensity. Removal of internal standard peaks and any known false positive peaks (including noise, column bleed and derivatized reagent peaks), was conducted, followed by peak pooling and redundancy elimination. At the same time, metabolites were identified using database such as the Human Metabolome Database (http://www.hmdb.ca/, accessed on 21 June 2022), Metlin (https://metlin.scrippss.edu/, accessed on 22 June 2022) and Majorbio Database. The processed data were then uploaded to the Majorbio cloud platform (https://cloud.majorbio.com, accessed on 22 June 2022) for subsequent data analysis.

### 2.6. Gene Expression Analyses in White Adipose Tissue

Total RNA was extracted from white adipose tissues with RNAiso Plus reagent (Takara, cat#9108, Beijing, China). The quality and concentration of total RNA were measured with a spectrophotometer (NanoDrop 2000, NanoDrop Technologies, LLC, Wilmington, DE, USA). Additionally, 200 ng/uL total RNA was subjected to reverse transcription using a HiScript III All-in-one RT SuperMix Perfect for qPCR Kit (Vazyme, Nanjing, China). Real-time qPCR was performed using a ChamQ Universal SYBR qPCR Master Mix kit (Vazyme, Nanjing, China) and ABI QuantStu-dio™ 6 Flex Real-time PCR System (ABI, Foster, CA, USA). The expression of target genes was normalized to the expression of *GADPH* and shown as a fold change relative to the control group based on the 2^−ΔΔCt^ method. The primer sequences are shown in Appendix A.

### 2.7. Metabolite Profiling of Cecal Contents Using GC-MS

A precise measurement of 200 mg of cecal contents was homogenized in 1 mL of water (using the Biological Sample Preparation System, Life Real Co., Ltd., Hangzhou, China) for about 3 min. The extraction of metabolites involved the addition of 0.15 mL 50% (*w*/*w*) H_2_SO_4_ and 1.6 mL of diethyl ether. The samples were then subjected to incubation in ice-cold water for 30 min, followed bt centrifugation at 5000× *g* for 15 min at 4 °C. The supernatant organic phase was analyzed using Agilent 7890B-7000B GC-MS (Agilent Technologies, Santa Clara, CA, USA), equipped with an HB-5 ms capillary column (30 m × 0.25 mm × 0.25 µm) (Agilent Technologies). The injector, ion source, quadrupole and GC/MS interface temperature were 260, 230, 150 and 280 °C, respectively. The flow rate of helium carrier gas was kept at 1 mL/min. Then, 1 µL of the derivatized sample was injected with a 3 min of solvent delay time and split ratio of 10:1. The initial column temperature was 40 °C, held for 2 min; then, the temperature ramped to 150 °C at the rate of 15 °C/min, was held 1 min, and then finally increased to 300 °C at the rate of 30 °C/min; and then, this temperature was maintained for 5 min. Ionization was carried out in the electron impact mode at 70 eV. The MS data were acquired in full scan mode from *m*/*z* 40 to 400 with an acquisition frequency of 12.8 scans per second. The identification of compounds was confirmed by searching in the reference library and comparing the retention time and corresponding MS spectra. 

### 2.8. Statistical Analyses

The results were presented as mean value along with standard deviation. Multiple group comparisons were assessed using one-way analysis of variance (ANOVA) followed by Tukey’s post hoc test (SPSS26, Inc., Chicago, IL, USA). Significant differences among means are indicated by different letters (*p* < 0.05). The representation of statistical significance is designated as * *p* < 0.05, ** *p* < 0.01, *** *p* < 0.001. The sample sizes and statistical tests employed are also indicated in the respective figure legends.

## 3. Results

### 3.1. PPE Ameliorated HFD-Induced Hepatic Steatosis in Rats

Our study aimed to delve into the intricate mechanisms that underscore the hepatoprotective effects of PPE in animal model subject to a high-fat diet, as depicted in the feeding schedule illustrated in Figure 1A. Initially, we concentrated on the effects of PPE supplementation on the body weight of rats with NAFLD. Remarkably, both low doses and high doses of PPE exhibited the capacity to curtail the body weight gain at sacrifice, when compared with rats exclusively fed a high-fat diet (Figure 1B), The reduction in body weight gain was particularly noteworthy in the high-dose PPE-supplemented group. As hypothesized, the serum lipid indexes were notably elevated in rats subject to HFD for 10 weeks, as compared with those fed a chow diet. Strikingly, upon PPE supplementation in the content if HFD, there was a significant decrease in the levels of triglycerides (TG), total cholesterol (TC) and low-density lipoprotein cholesterol (LDL-C) in the serum. Furthermore, the levels of high-density lipoprotein cholesterol (HDL-C) were augmented (Figure 1C–F).

Prolonged consumption of a high-fat diet tends to cause the accumulation of lipid droplets within the liver, resulting in the manifestation of NAFLD. We employed oil red O staining to visualize the accumulation of lipid droplets in liver tissue. As shown in Figure 2A, the liver sections of rats from the high-fat diet group predominantly exhibited orange-red lipid droplet accumulation, indicating diet-induced hepatic steatosis. Interestingly, the introduction of a low-dose PPE intervention resulted in a reduction in these orange-red lipid droplets, whereas a high-dose PPE intervention almost restored the quantity of lipid droplets to levels akin to the chow diet control group. These findings underscored the substantial inhibitory effect of PPE on liver steatosis in rats. The addition of PPEs to the diet effectively mitigated the buildup of lipid droplets in the liver tissue, subsequently revitalizing the liver’s metabolic homeostasis. Statistical analysis of the lipid droplet area in liver tissue, depicted in Figure 2B, reaffirmed the potency of PPE in reducing the accumulation of lipid droplets, with its efficacy being dosage-dependent.

### 3.2. PPE Extensively Modulated Hepatic Lipid Profile

In pursuit of a deeper understanding of PPEs’ impacts on lipid metabolism within the fatty liver context, we conducted a lipidomic analysis using UHPLC−Q Exactive HF−X technology. This assessment aimed to discern intrahepatic lipid profile alterations among the HFD control group, Chow control group and PPE_E group, as depicted in Appendix A. A rigorous analysis of retention time and peak area variations in QC samples yielded a relative standard deviation (RSD) of components below 0.3, coupled with a cumulative proportion of peaks surpassing 70%, thus affirming the data qualification (Figure 3A,B). Leveraging precise *m*/*z*, retention time and MS/MS, we successfully identified 788 lipids in positive ion mode and 507 lipids in negative ion mode, culminating a total of 1295 identified lipids spanning 13 distinct lipid subclasses (Figure 3C). Notably, “Triradylcglycerols” emerged as the most numerous lipid subclass.

Partial least squares discriminant analysis (PLS−DA) underscored distinctive intrahepatic lipid profiles between the HFD, chow diet and PPE intervention groups, evident in both the positive and negative detection modes of the LC−MS/MS lipidomic analysis (Figure 4A,B). Remarkably high R^2^Y and Q^2^ values (close to 1) of 0.992 and 0.985, respectively, for positive mode and 0.832 and 0.778 (greater than 0.5) for negative mode affirmed the stability, reliability and predictive prowess of the models. Permutation testing, with Y-axis intercepts of Q2 below 0.05 (Figure 4C,D) indicated that over-fitting of the model was not a concern.

In order to delineate the differential lipid components between two groups (PPE_H vs. HFD, HFD vs. Chow), we performed a pairwise orthogonal partial least squares discriminant analysis (OPLS-DA) on the lipidomic data. The screening criteria encompassed VIP in the OPLS-DA model (VIP > 2.5) and the *p*-value of the Student’s *t*-test (*p* < 0.05). As a result, 13 lipid components exibited significant differences in the PPE_H vs. HFD group, including Cardiolipin (CL), Dilysocardiolipin (DLCL), Ganglioside GM3 (GM3), Lysocardiolipin (MLCL), Phosphatidylinositol (PI), Diacylglycerol (DG), Phosphatidylethanolamine (PE) and TG. Detailed information of the differential lipid components is shown in Appendix A, including the molecular formula, retention time, *m*/*z*, substance identification and difference significance score of the lipid components. Furthermore, in the HFD vs. CON group, nine lipid components were identified as differentially significant, including CL, PE and TG. Detailed data on these differential lipid components are presented in Appendix A.

We used a Z−score scatter plot to show the distribution of differential metabolites across the two sets (Figure 5A,B). It can be seen from the diagram that the deposition of triglyceride (TG) and diacylglycerol (DG) within the liver was significantly reduced under PPE intervention. In contrast, the TG content significantly increased in the high-fat diet group compared to the chow diet group. The contents of PE, LPM and LPG within the liver of rats for the high-fat diet group were significantly lower than those in the chow diet group. After PPE supplementation, the content of glycerophosphate PE was significantly higher than that in the high-fat diet group, but the content of PI was decreased in the PPE group. The content of CL in the high-fat diet group was higher than that in the chow diet group, and the content was significantly reduced after the addition of PPE. Additionally, the PPE intervention group exhibited lower contents of MLCL, DLCL and GM3 in comparison with the HFD group. Therefore, the inclusion of PPE in the diet reversed TG accumulation in the liver of NAFLD rats caused by prolonged high-fat diet consumption, leading to a significant reduction in content, along with an elevation in beneficial fatty acid PE content. By altering the lipid composition within the liver of NAFLD rats, PPE exhibited a potent capacity to alleviate liver injury.

For a comprehensive understanding, we conducted a KEGG metabolic pathway enrichment analysis using differential metabolites of PPE_H vs. HFD as the metabolic set and used the hypergeometric distribution algorithm to obtain the significantly enriched metabolic pathways. This revealed the profound impact of PPE intervention on the pathways related to glucose and lipid metabolism. The pivotal pathways encompassed lipolysis in adipocytes, insulin resistance, thermogenesis, fat digestion and absorption, insulin secretion, cholesterol metabolism, the phospholipase D signaling pathway, the adipocytokine signaling pathway, the sphingolipid signaling pathway and the MAPK signaling pathway (Figure 6).

In essence, our findings corroborate the significant mitigation effect of PPE intervention on NAFLD-related liver steatosis, mainly by promoting lipolysis and thermogenesis, enhancing insulin secretion, alleviating insulin resistance and regulating related signaling pathways. PPE intervention effectively counteracts lipid droplet accumulation stemming from prolonged high-fat diet consumption, thus offering promise for NAFLD mitigation.

### 3.3. PPE Regulated Expression of the Lipolysis Gene in White Adipose Tissue and Strengthened Mobilization from Fat

Hormone-sensitive lipase (HSL), a pivotal enzyme governing triglyceride breakdown in adipose tissue, holds a key role in adipose lipid metabolism regulation. Our findings demonstrate that PPE intervention exerts a promotive effect on the expression of *HSL* gene. Notably, high-dose PPE intervention leads to a significant *HSL* up-expression, thereby promoting lipid decomposition within adipose tissue (Figure 7A). Adipose triglyceride lipase (ATGL) serves as a principal enzyme initiating fat mobilization in adipose tissue. ATGL is responsible for the specific hydrolysis of the initial ester bond within triglyceride, constituting a rate-limiting step in triglyceride hydrolysis. Notably, mutations in the *ATGL* gene disrupt the body’s intrinsic energy metabolism balance, promoting ectopic deposition of excess fat in various tissues and fostering metabolic syndrome. Our observations reveal that compared with the chow diet control group, *ATGL* expression diminishes in the high-fat diet group, while the high-dose PPE intervention significantly increased the *ATGL* expression level (Figure 7B). This finding validates that PPE actively advances adipose tissue liposis. Uncoupling protein 1 (UCP1) and uncoupling protein 2 (UCP2) assume pivotal roles in the energy metabolism of adipose tissue. The up−regulation of *UCP1* and *UCP2* genes associated with thermogenesis and the preservation of energy equilibrium. Conversely, their down-regulation leads to lipid accumulation. In this study, we found that high-dose PPE intervention significantly elevates *UCP1* and *UCP2* mRNA expression. This signifies that PPE not only spurs lipid breakdown but also curbs lipid synthesis, thereby thwarting lipid accumulation by promoting the expression of *UCP1* and *UCP2*.

### 3.4. PPE Affected Intestinal Metabolites: Long-Chain Fatty Acids and Short-Chain Fatty Acids

We delved into the impact of PPE intervention on the levels of long-chain fatty acids (LCFAs) and short-chain fatty acids (SCFAs) in cecal contents by conducting gas chromatography–mass spectrometry. Through database comparisons and peak integration, we discerned the LCFAs and SCFAs with significant content variations. Detailed information including name, structural formula, retention time and content abundance in each group can be found in Table 1 and Table 2. Distinct lowercase letters signify differences between groups (*p* < 0.05).

Our findings reveal that the prolonged consumption of a high-fat diet leads to an increase in the contents of seven LCFAs in rat cecal contents. However, after PPE intervention, the levels of these seven LCFAs substantially decreased. Moreover, the contents of five SCFAs within the cecal contents of NAFLD rats display a noteworthy increase following PPE intervention (compared with the high-fat diet control group). This substantiates the regulatory effect of PPE intervention on lipid metabolism in NAFLD rats. This phenomenon points toward PPEs’ potential to facilitate the conversion of LCFAs to SCFAs, thereby contributing to metabolic modulation.

## 4. Discussion

This study underscores the efficacy of PPE intervention in ameliorating NAFLD. Both low and high doses of PPE intervention effectively curbed body weight gain and improved key blood lipid parameters, demonstrating a dose-dependent response. Notably, PPE intervention exerted a hepatoprotective effect by ameliorating lipid droplet accumulation within the livers of NAFLD rats. Furthermore, PPE induced transformative shifts in liver lipid composition, thereby mitigating liver injury. Through its multifaceted impact, PPEs significantly alleviated NAFLD liver steatosis, primarily by augmenting lipolysis and thermogenesis, encouraging thermogenesis, including insulin secretion, alleviating insulin resistance and modulating related signaling pathways. Within adipose tissue, PPE intervention fostered lipid decomposition, mobilization and synthesis limitation, thus preventing lipid accumulation. Additionally, PPE intervention influenced intestinal metabolites, reducing the content of LCFAs while elevating the content of SCFAs. In general, PPE nutritional interventions were seen to alter lipid composition in the liver, to enhance lipid metabolism and to regulate intestinal metabolites for mitigation of NAFLD.

**Heterogeneous etiology for NAFLD and PPE effects.** NAFLD represents a widespread chronic liver ailment characterized by heterogeneous etiology involving metabolism, inflammation, genetics, environment and gut microbiota. Presently, lifestyle interventions, such as dietary adjustments and exercise, stand as the primary treatments for NAFLD. Experimental evidence points to the therapeutic potential of natural foods for NAFLD [11]. The natural foods that have been reported to alleviate NAFLD include artichoke leaf [12], lycium barbarum polysaccharide [13], red yeast rice [14] and garlic [15,16]. These foods exhibit activities akin to PPE, spanning antioxidant, anti-inflammatory, ameliorate lipid metabolism improvement and gut microbiota modulation.

Hepatic steatosis marks NAFLD’s initial stage of development and progression [17]. The liver sections stained with oil red underscore the extensive lipid droplet accumulation within hepatocytes of NAFLD rats. Such fatty degeneration renders hepatocytes susceptible to injury, magnifying lipid metabolism disturbances and inflammation and inducing liver damage [18]. The ability of PPEs to scavenge lipid droplets in the liver cell protects hepatocytes from further harm. Intraliver fat buildup epitomizes core NAFLD hallmarks [19], stemming from heightened fatty acid uptake, augmented de novo lipogenesis, hindered fatty acid oxidation and exacerbated insulin resistance [20]. Insulin resistance sparks the de novo synthesis of fatty acids in the liver, activating sterol regulatory element-binding protein-1c, a transcriptional regulator of adipogenic genes [21]. These synthesized and dietary fatty acids are either subjected to β-oxidation or esterification with glycerides to form triglycerides. Consequently, these triglycerides are stored within hepatocytes as lipid droplets or channeled into LDL-C [19]. Notably, PPE intervention led to a substantial reduction in LDL-C levels within NAFLD rats’ serum, showcasing its ability to counteract LDL-C elevation in NAFLD.

**Emerging Insights from Lipidomics and Lipid Metabolism.** Recent lipidomics studies have unveiled significant alterations in fatty acid patterns and phospholipid composition within liver samples from NAFLD patients, underscoring lipid metabolism disorders as pivotal in the pathogenesis and progression of NAFLD [22,23]. Our study corroborated these findings, revealing that PPE intervention notably curtailed the deposition of neutral lipids, such as TG and DG in the liver. These neutral lipids, extensively implicated in prior research, have exhibited pronounced elevation in the liver biopsy of NAFLD patients [23,24]. Additionally, the contents of phospholipids such as phosphatidylethanolamine (PE), lysophosphatidylcholine (LPM) and lysophosphatidylglycerol (LPG) were considerably lower in the liver of NAFLD rats compared with the chow diet group, with PE levels significantly increasing following PPE supplementation. Phosphatidylcholine (PC), a major component of cell membrane lipids and abundant phospholipids in mammals, demonstrated decreased levels in liver samples of NAFLD patients [25]. In the metabolic context, PC synthesis stems from dietary choline through the cytidine 5′-diphosphate CDP-choline pathway [26]. In hepatocytes, phosphatidylethanolamine N-methyltransferase enzyme catalyzes the conversion of PE to PC, constituting up to 30% of PC synthesis [27]. The synthesis of PE itself proceeds through the CDP-ethanolamine pathway and phosphatidylserine (PS) decarboxylation. Notably, liver PE content has been shown to significantly decline in NAFLD subjects. We infer that PPE supplements likely counteract insulin resistance and mitigate NAFLD by modulating neutral lipid metabolism and phospholipid metabolism within the liver.

The contemporary understanding of NAFLD positions it as a metabolically associated fatty liver disease, with its pathophysiology intertwined with abnormal communication within the adipose tissue–liver axis [28]. In this vein, inflammation and metabolic disturbances within obesity-linked white adipose tissue trigger liver insulin resistance, ectopic lipid accumulation and eventually liver maladies [29]. Our findings corroborate this paradigm, elucidating that PPE intervention engenders regulatory effects on white adipose tissue metabolism. Specifically, PPE augmented the expression of *HSL* and *ATGL* genes, promoting lipid mobilization. Additionally, the up-regulation of *UCP1* and *UCP2* genes was observed, concomitantly limiting fat synthesis and thwarting lipid accumulation. This collective evidence suggests that the effect of PPE on improving NAFLD may stem from its dual influence on restraining lipogenesis while inducing fatty acid oxidation induction within white adipose tissue.

**Emergence of Dietary and Intestinal Factors in NAFLD Pathogenesis.** In recent times, the role of dietary constituents and intestinal metabolites in the development of NAFLD has gained increasing attention [30]. The impact of intestinal metabolites extends across diverse aspects of NAFLD progression, influencing energy balance, intestinal permeability, choline metabolism and SCFA production. SCFAs are produced by the fermentation of carbohydrates by the gut flora, serve as vital energy sources of intestinal epithelial cells and can also reach the liver via the hepatic vein. In the liver, they partake in various biosynthetic pathways, such as gluconeogenesis and lipid biosynthesis [31]. For example, butyrate has been linked to increased glucose production and elevated expression of gluconeogenesis-related genes in primary hepatocytes of rats. In addition, SCFAs engage with G protein-coupled receptors, including GPR41 and GPR43, influencing lipid metabolism. In an obesity model, butyrate administration mitigated hepatic steatosis by modulating GPR41 and GPR43 expression [32]. Therefore, PPE intervention’s augmentation of five SCFAs (acetic acid, propionic acid, 2-methylpropionic acid, butyric acid and valeric acid) in intestinal metabolites underscores its favorable impact on liver steatosis, glucose metabolism and lipid homeostasis in NAFLD rats. In contrast, previous research has highlighted a positive correlation between the abundance of long-chain saturated fatty acids in fecal metabolites and the severity of metabolic disorders such as obesity, NAFLD and type 2 diabetes mellitus (T2DM) [33]. It is plausible that long-chain fatty acids intersect with the lipopolysaccharide metabolic pathway, potentially exacerbating intestinal inflammation and consequently the development of NAFLD. Intriguingly, PPEs appear to counteract this trend by diminishing the content of long-chain saturated fatty acid in the intestinal metabolites of NAFLD rats, potentially intervening in the cascade of events leading to intestinal inflammation and NAFLD progression.

## 5. Conclusions

In this study, we harnessed the potential of *Arthrospira platensis* PPEs as a therapeutic tool against HFD-induced hepatic steatosis. PPEs demonstrated the capacity to ameliorate NAFLD, with their hepatoprotective effects stemming from multifaceted mechanisms. PPEs exert an influence on hepatic lipid metabolism, modulating both neutral lipid and phospholipid dynamics within the liver. Furthermore, PPEs’ roles in white adipose tissue are pivotal, as they orchestrate the expression of lipolysis genes, culminating in augmented fat mobilization. Meanwhile, PPEs exert an impact on intestinal metabolites, augmenting SCFAs, which regulate diverse metabolic pathways. These revelations offer a fresh perspective on PPEs’ hepatoprotective potential, furnishing a theoretical foundation for their utilization as a functional component to counteract NAFLD. This study underscores the multifunctional mechanisms of PPEs in mitigating NAFLD, thereby paving the way for innovative interventions in the realm of non-alcoholic fatty liver ailment.

## Figures and Tables

**Figure 1 nutrients-15-04573-f001:**
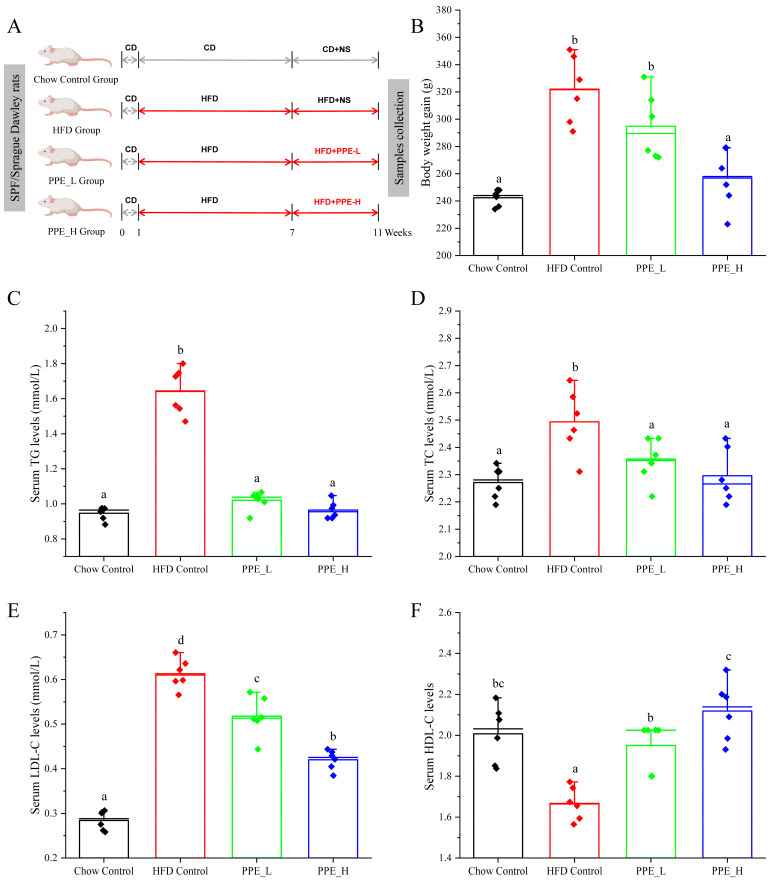
PPE administration mitigated HFD-induced overweight and hyperlipidemia. (**A**) Feeding schedule; (**B**) body weight gain; (**C**) TG, (**D**) TC, (**E**) LDL−C and (**F**) HDL−C levels in serum. Data are presented by the mean ± SD (*n* = 6). Different letters indicate significant difference (*p* < 0.05) between groups. CD: chow diet; HFD: high-fat diet; NS: normal saline; PPE: phycobiliprotein bioactive peptide extract; Chow Control: chow diet control group; HFD Control: high fat diet control group; PPE_L: high fat diet with low−dose PPE group; PPE_H: high fat diet with high−dose PPE group; TG: triglyceride; TC: total cholesterol; LDL−C: low−density lipoprotein cholesterol; HDL−C: high−density lipoprotein cholesterol.

**Figure 2 nutrients-15-04573-f002:**
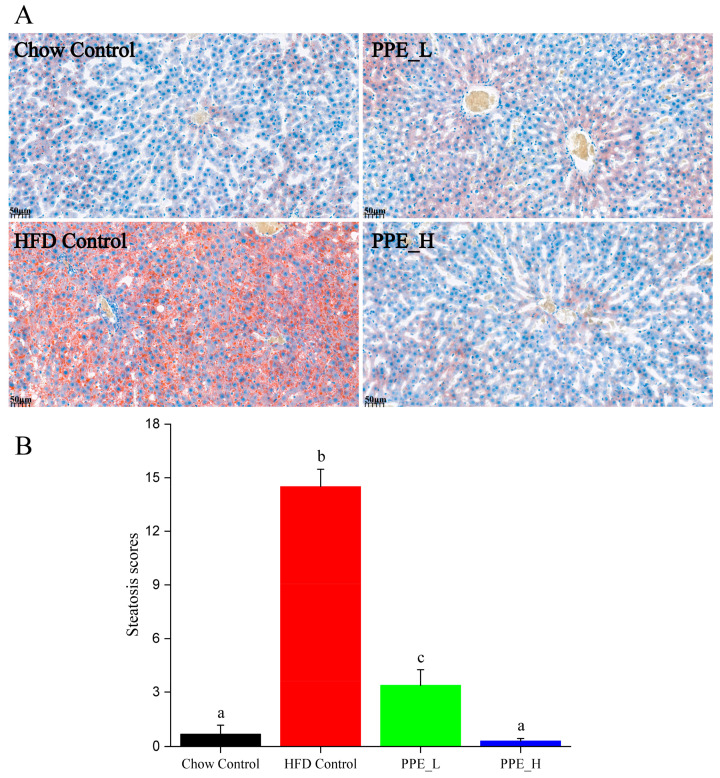
PPE alleviated HFD induced hepatic steatosis in rats. (**A**) Oil red O staining of liver tissues (scale bar: 50 µm). (**B**) Steatosis scores of lipid droplet in fatty liver tissues. Data are presented as the mean ± SD (*n* = 6). Different letters indicate significant difference (*p* < 0.05) between groups. PPE: phycobiliprotein bioactive peptide extract; Chow Control: chow diet control group; HFD Control: high fat diet control group; PPE_L: high fat diet with low−dose PPE group; PPE_H: high fat diet with high−dose PPE group.

**Figure 3 nutrients-15-04573-f003:**
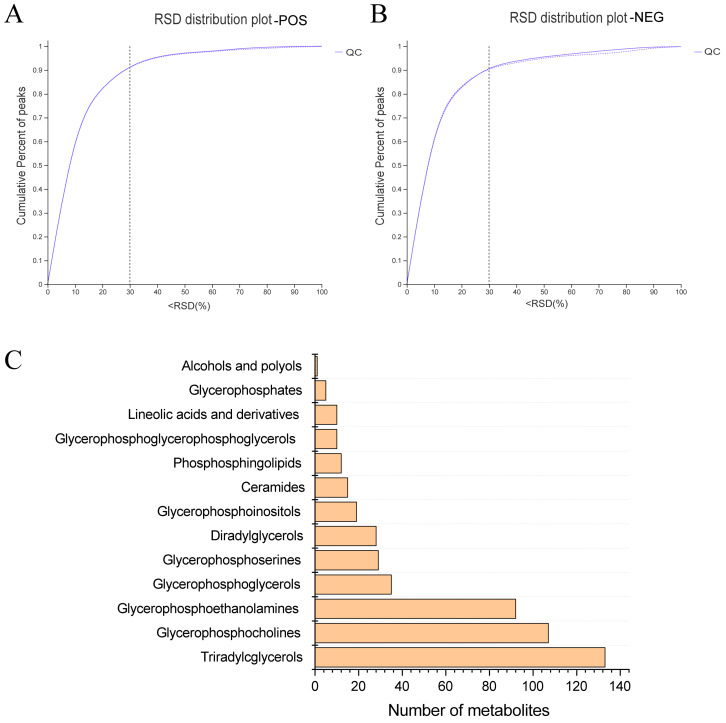
Overview of lipidomics data. (**A**) The evaluation diagram of QC sample in positive ion mode. (**B**) The evaluation diagram of QC sample in negative ion mode. (**C**) Classification of liver metabolites in human metabolome database. RSD: relative standard deviation; POS: positive ion pattern; NEG: negative ion pattern; QC: quality control sample.

**Figure 4 nutrients-15-04573-f004:**
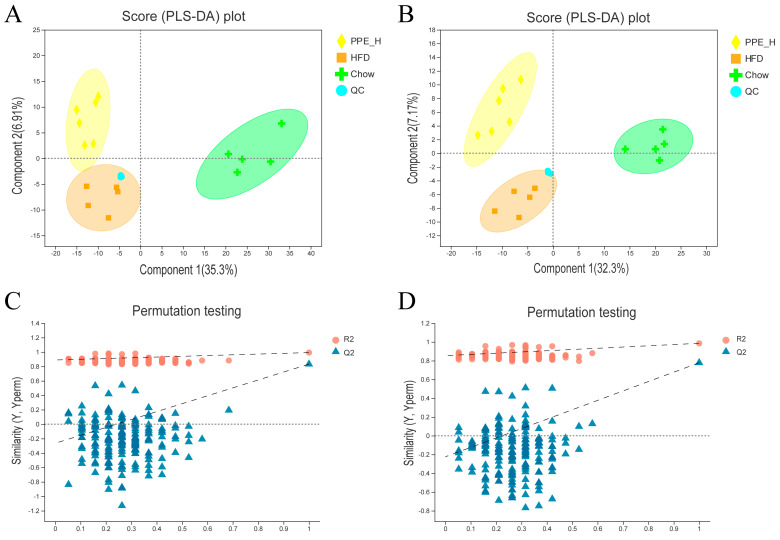
Comparison analysis of metabolite profiling. (**A**) PLS−DA score plot of the samples in the positive ion detection modes. (**B**) PLS−DA score plot of the samples in negative ion detection modes. (**C**) PLS−DA model validation in the positive ion detection modes. (**D**) PLS−DA model validation in negative ion detection modes. PPE: phycobiliprotein bioactive peptides extract; Chow Control: chow diet control group; HFD Control: high fat diet control group; PPE_L: high fat diet with low−dose PPE group; PPE_H: high fat diet with high−dose PPE group; PLS−DA: partial least squares discriminant analysis.

**Figure 5 nutrients-15-04573-f005:**
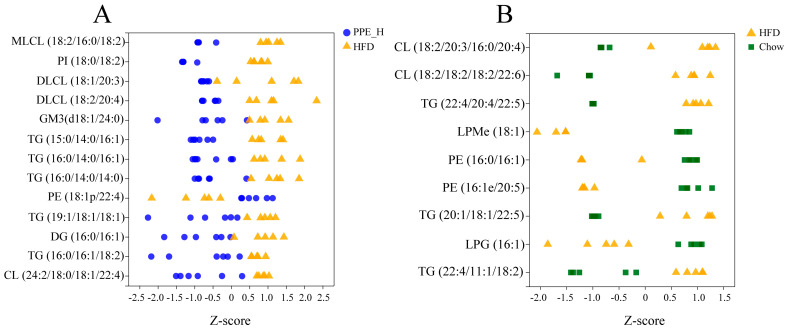
The Z−score scatter plot of differential lipid metabolites. (**A**) Abundance distribution of differential lipid metabolites between PPE_H and HFD groups. (**B**) Abundance distribution of differential lipid metabolites between HFD and Chow diet groups. PPE: phycobiliprotein bioactive peptides extract; Chow Control: chow diet control group; HFD Control: high fat diet control group; PPE_H: high fat diet with high−dose PPE group.

**Figure 6 nutrients-15-04573-f006:**
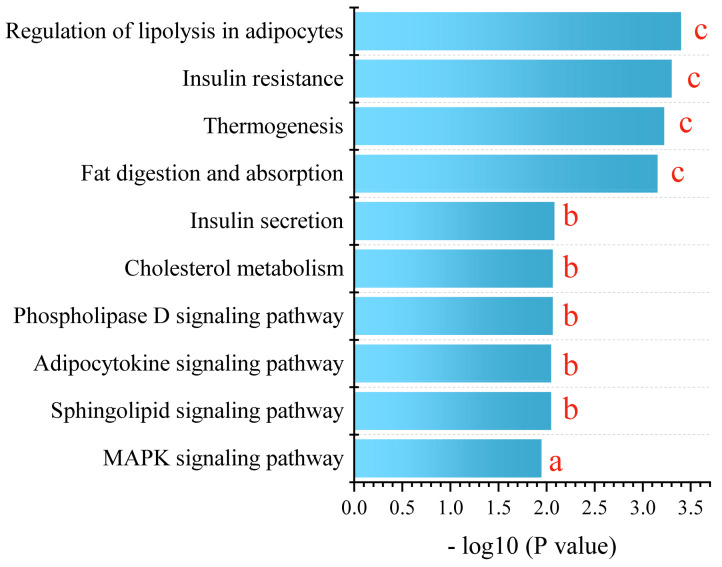
KEGG enrichment analysis between PPE_H and HFD. a represents *p* < 0.05, b represents *p* < 0.01 and c represents *p* < 0.001. (*n* = 6 rats in each group.) PPE: phycobiliprotein bioactive peptides extract; PPE_H: high fat diet with high−dose PPE group; HFD: high fat diet control group.

**Figure 7 nutrients-15-04573-f007:**
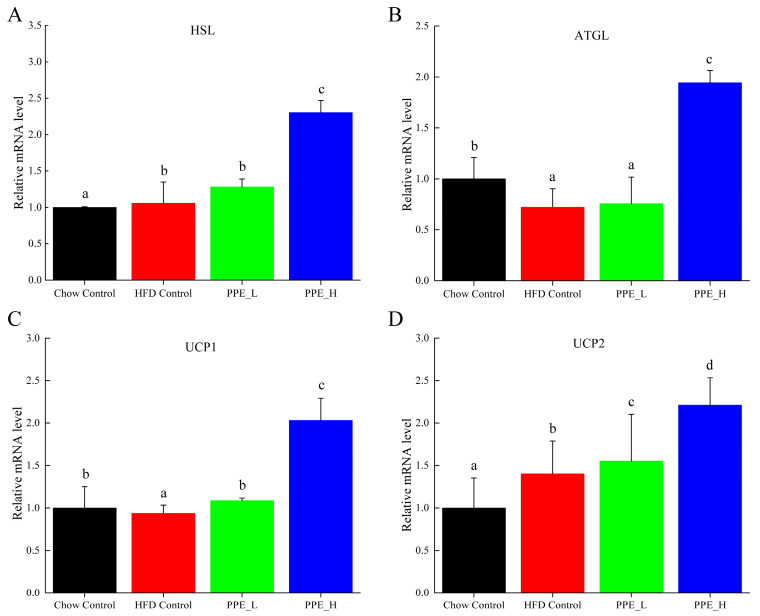
Effects of PPE supplementation on the gene expression of fat mobilization in white adipose tissue. (**A**) The mRNA expression level of *HSL* in white adipose tissue. (**B**) The mRNA expression level of *ATGL* in white adipose tissue. (**C**) The mRNA expression level of *UCP1* in white adipose tissue. (**D**) The mRNA expression level of *UCP2* in white adipose tissue. Data are presented as the mean ± SD (*n* = 6). Different letters indicate significant difference (*p* < 0.05) between groups. Chow Control: chow diet control group; HFD Control: high fat diet control group; PPE_L: high-fat diet with low−dose PPE group; PPE_H: high-fat diet with high−dose PPE group; HSL: hormone-sensitive lipase; ATGL: adipose triglyceride lipase; UCP1: uncoupling protein 1; UCP2: uncoupling protein 2.

**Table 1 nutrients-15-04573-t001:** Qualitative and quantitative comparison of long chain fatty acids in cecal contents. Data are presented as the mean ± SD (*n* = 6). Different letters indicate significant difference (*p* < 0.05) between groups.

NO.	Compounds Name	Structural Formula	RT (min)	Chow Control	HFD Control	PPE_L	PPE_H
1	9,12-Octadecadienoic acid (Z,Z)-	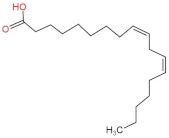	51.37	(1.24 ± 0.07) × 10^7 a^	(5.71 ± 0.16) × 10^7 d^	(3.17 ± 0.17) × 10^7 c^	(2.59 ± 0.16) × 10^7 b^
2	Oleic Acid	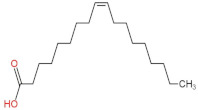	48.497	(3.52 ± 0.24) × 10^7 a^	(3.18 ± 0.61) × 10^8 d^	(2.41 ± 0.30) × 10^8 c^	(1.28 ± 0.10) × 10^8 b^
3	Octadecanoic acid	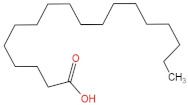	47.088	(3.65 ± 0.17) × 10^7 a^	(3.64 ± 0.31) × 10^8 c^	(2.02 ± 0.48) × 10^8 b^	(1.77 ± 0.22) × 10^8 b^
4	Heptadecanoic acid	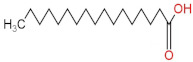	42.723	(1.62 ± 0.28) × 10^6 a^	(8.51 ± 0.14) × 10^6 d^	(7.16 ± 0.33) × 10^6 c^	(5.08 ± 0.58) × 10^6 b^
5	n-Hexadecanoic acid	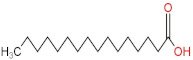	39.389	(1.57 ± 0.27) × 10^8 a^	(2.58 ± 0.29) × 10^9 d^	(1.89 ± 0.16) × 10^9 c^	(1.60 ± 0.19) × 10^9 b^
6	Pentadecanoic acid	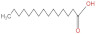	36.699	(3.98 ± 0.20) × 10^6 a^	(1.16 ± 0.27) × 10^7 c^	(7.67 ± 0.50) × 10^6 b^	(5.80 ± 0.45) × 10^6 ab^
7	Tetradecanoic acid	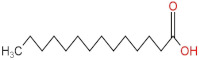	34.56	(2.72 ± 0.23) × 10^6 a^	(3.33 ± 0.37) × 10^7 d^	(2.04 ± 0.34) × 10^7 c^	(1.54 ± 0.19) × 10^7 b^

**Table 2 nutrients-15-04573-t002:** Qualitative and quantitative comparison of short chain fatty acids in cecal contents. Data are presented as the mean ± SD (*n* = 6). Different letters indicate significant difference (*p* < 0.05) between groups.

NO.	Compounds Name	Structural Formula	RT (min)	Chow Control	HFD Control	PPE_L	PPE_H
1	Pentanoic acid	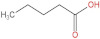	16.996	(2.01 ± 0.55) × 10^6 a^	(1.85 ± 0.47) × 10^6 a^	(3.77 ± 0.66) × 10^6 b^	(3.29 ± 0.65) × 10^6 b^
2	Butanoic acid	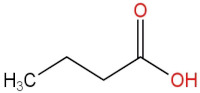	14.63	(5.32 ± 0.40) × 10^6 a^	(5.83 ± 0.67) × 10^6 a^	(1.23 ± 0.17) × 10^7 c^	(8.95 ± 0.74) × 10^6 b^
3	Propanoic acid, 2-methyl-	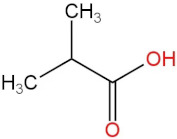	13.27	(9.96 ± 0.67) × 10^5 a^	(1.04 ± 0.10) × 10^6 a^	(1.75 ± 0.50) × 10^6 b^	(1.33 ± 0.21) × 10^6 a^
4	Propanoic acid	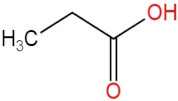	12.618	(4.46 ± 0.40) × 10^6 a^	(6.94 ± 0.35) × 10^6 b^	(9.59 ± 0.70) × 10^6 c^	(7.32 ± 0.70) × 10^6 b^
5	Acetic acid	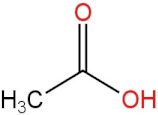	10.66	(4.38 ± 0.36) × 10^6 a^	(5.79 ± 0.66) × 10^6 b^	(8.68 ± 0.62) × 10^6 d^	(6.60 ± 0.43) × 10^6 c^

## Data Availability

All data generated or analyzed during this study are included in this published article and its Appendix A files.

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
