# Peer review of "Phycobiliprotein Peptide Extracts from Arthrospira platensis Ameliorate Nonalcoholic Fatty Liver Disease by Modulating Hepatic Lipid Profile and Strengthening Fat Mobilization"

_nutrients, 2023, doi:10.3390/nu15214573_

Round 1
Reviewer 1 Report
Comments and Suggestions for Authors
In this original manuscript, the authors aimed to investigate the impact of phycobiliproteins peptide extracts in high-fat-induced nonalcoholic fatty liver disease (NAFLD) and the underlying mechanism. There are several concerns should be addressed before it can be considered for publication.
1. Some data/results/conclusions have been published in a similar paper by the author groups.
2. The mechanism of PPE ameliorated HFD-induced hepatic steatosis in rat is unclear.
3. A lot of data is presented but they are is not well organized and linked together and support the study.
4. The significant labels and graph error bar should be checked. The significant labels and graphs are confused. The significant labels should be consistent (use letters or symbols, not both). The Figure legends are not detailed.
5. The HFD model should be checked. In Figure 2A, it is barely to see hepatocyte ballooning in HFD group. Higher magnification and H&E image are requested. In Figure 2B, how the steatosis scores were assessed. Did authors check liver enzyme in rat serum?
6. Did liver/body weight ratio change after PPE treatment? Does PPE alter rat food intake? Did the authors record food intake for the rat?
7. The authors stated that “PPE regulated expression of the lipolysis gene in white adipose tissue”. Does PPE alter white adipose tissue weight. Does PPE regulate brown fat metabolism?
8. Did author treat hepatocytes/adipocytes in vitro with PPE to confirm the metabolic pathway.
Comments on the Quality of English Languageminor grammatical errors
Author Response
Dear Reviewer:
We provide a point-by-point response to the comments and upload it as a Word file. Please see the attachment.

Reviewer 2 Report
Comments and Suggestions for Authors
Dear authors, please find below my comments and suggestions:
L93: rats are housed individually. it is very surprising that the rats are not housed with others. indeed, individual cages can induce stress. Please justify that and explain how signs of stress have been followed.
L99-101: please describe the quality of used fat.
L109-110: euthanasy by ether inhalation is not accepted. Please explain why this method is used and how the authors were sure that the animals were well asleep.
L243: scale bars are not reported in the figure 2A
Figure 7: it is reported that there is no significant difference between the PPE_L group and the PPE_H while there is one with HFD control. Please verify this point.
Author Response

(The authors gave the same response as above.)
